# Integrating Markov processes with structural causal modeling enables counterfactual inference in complex systems

**Robert Ness**
Gamalon Inc.
robert.ness@gamalon.com

**Kaushal Paneri**
Northeastern University
kaushalpaneri@gmail.com

**Olga Vitek**
Northeastern University
o.vitek@northeastern.edu

## Abstract

This manuscript contributes a general and practical framework for casting a Markov process model of a system at equilibrium as a structural causal model, and carrying out counterfactual inference. Markov processes mathematically describe the mechanisms in the system, and predict the system's equilibrium behavior upon intervention, but do not support counterfactual inference. In contrast, structural causal models support counterfactual inference, but do not identify the mechanisms. This manuscript leverages the benefits of both approaches. We define the structural causal models in terms of the parameters and the equilibrium dynamics of the Markov process models, and counterfactual inference flows from these settings. The proposed approach alleviates the identifiability drawback of the structural causal models, in that the counterfactual inference is consistent with the counterfactual trajectories simulated from the Markov process model. We showcase the benefits of this framework in case studies of complex biomolecular systems with nonlinear dynamics. We illustrate that, in presence of Markov process model misspecification, counterfactual inference leverages prior data, and therefore estimates the outcome of an intervention more accurately than a direct simulation.

## 1 Introduction

Many complex systems contain discrete components that interact in continuous time, and maintain interactions that are stochastic, dynamic, and governed by natural laws. For example, molecular systems biology studies molecules (e.g., gene products, proteins) in a living cell that interact according to biochemical laws. An important aspect of studying these systems is predicting the equilibrium behavior of the system upon an intervention, and selecting high-value interventions. For example, we may want to predict the effect of a drug intervention on a new equilibrium of gene expression [1, 27]. The intervention may have a high value if reduces the expression of a specific gene, while minimizing changes to the other genes.

Recent work in the reinforcement learning community has highlighted the utility of *counterfactual policy evaluation* for evaluating and comparing interventions. Counterfactual policy evaluation uses data from past experimental interventions to ask whether a higher value could have been achieved under an alternative intervention [7, 16, 8, 19]. Counterfactual inference answers this question by predicting the outcome of the alternative intervention, conditional on the outcome of the intervention for which the data were observed [7, 21].

Predicting the outcome of an intervention requires us to model the system. In particular, *discrete-state continuous-time Markov process models* unambiguously describe the changes of system components across all the system states (i.e., not only at equilibrium) in term of hazard functions [11, 28]. A Markov process model predicts the equilibrium upon an intervention by applying the intervention to the initial conditions, performing multiple direct stochastic simulations to reach post-intervention equilibriums, and averaging over these equilibriums. Markov process modeling is one way of modeling complexity in biological systems, particularly in systems that are intrinsically stochastic [1]. The Markov process models are called stochastic kinetic models in this context.

Unfortunately, Markov process models do not support counterfactual inference. Moreover, it is often impossible to correctly specify a Markov process model of a complex system such as a biological system, where many aspects of the underlying mechanism are unknown. Direct simulations from an incorrectly specified model may incorrectly predict the outcomes of interventions.

An alternative class of models are *structural causal models* (SCMs). These probabilistic generative causal models are attractive, in that they enable both interventional and counterfactual inference [20]. Recent work used SCMs to model the transition functions in simple Markov decision process models and apply counterfactual policy evaluation to the decisions (i.e. interventions) at each time step [8, 19]. Unfortunately, these approaches require outcome data at each time point. This limits their use in situations where we are only interested in the outcome at equilibrium, and only collect data once the equilibrium is reached.

Defining SCM models at equilibrium directly is non-trivial, because multiple SCMs may be consistent with the equilibrium distribution of the system components upon an intervention, but provide contradictory answers to the same counterfactual query [20, 23]. Recent work [6, 18, 5] connected a broader class of dynamic models and SCMs, and established the conditions under which interventions in dynamic simulations correspond to SCM's predictions of equilibrium upon the interventions. However, researchers lack practical examples that leverage this connection, and combine the benefits of these two approaches for counterfactual inference.

This manuscript builds on these prior results, and contributes a general and practical framework for casting the equilibrium distribution Markov process model as an SCM model of equilibrium behavior. The SCMs are defined in terms of the structure and the hazard rates parameters of the Markov process model, and counterfactual inference flows from these settings. The proposed approach alleviates the identifiability drawback of the SCMs, in that their counterfactual inference is consistent with the counterfactual trajectories simulated from the Markov process model. We showcase the benefits of this approach in two studies of cell signal transduction with nonlinear dynamics. The first is a canonical model of the MAPK signaling pathway [17]. The second is a larger model that connects the MAPK pathway to stimulus from growth factors [3]. We illustrate that, when the underlying Markov process model is misspecified, counterfactual inference anchors intervention predictions to past observed data, and makes selection of interventions more robust to model misspecification.

## 2 Background

**Discrete-state continuous-time Markov process models** Discrete-state continuous-time Markov process models describe the temporal interactions between the system components in terms of abstract or physical processes, called *rate laws*, with real-valued parameters *rates* [11]. The rate laws determine *hazard functions*, which provide instantaneous probabilities of state transitions.

A *place invariant* is a set of system components with an invariant sum. A *minimal place invariant* can not be further reduced to smaller place invariants [9]. Define random variables $\mathbf{X}(t) = \{X_i(t) : i \in 1....J\}$ representing the states of $J$ minimal place invariant components in a Markov process model. We use capital letters to refer to random variables, lower case letters to refer to instances of random variables, normal font for a single variable, and boldface for a tuple of variables. Denote $P^{\mathbb{M}}(t)$ the probability distribution of $\mathbf{X}(t)$, and $P_{X_i}^{\mathbb{M}}(t)$ the marginal probability of $X_i(t)$. A Markov process model $\mathbb{M}$ is defined by *master equations*, i. e. a coupled set of ordinary differential equations that describe the rate of change of the probabilities of the states $\mathbf{X}(t)$ over time [29]:

$$\frac{dP_{X_i}^{\mathbb{M}}(t)}{dt} = h_i\left(t, \mathbf{v}_i, \mathbf{PA}_{\mathbb{M},i}(t)\right), \ X_i(0) = (x_0)_i \ \forall i \in J \tag{1}$$

The function $h_i$ is the hazard function that determines the probability of a state change between $X_i(t)$ and $X_i(s)$, $s > t$. Here $\mathbf{v}_i$ is a set of parameters of the rate laws, and $\mathbf{x}_0$ is an initial condition. $\mathbf{PA}_{\mathbb{M},i}(t) \subseteq \mathbf{X}(t) \setminus X_i(t)$ is the set of *parents* of variable $X_i(t)$, i.e. variables that regulate $X_i(t)$. Here we consider only Markov process models that converge to unique equilibrium stationary distributions. If equilibrium exists, then $\lim_{t \to \infty} \frac{dP_{X_i}^{\mathbb{M}}(t)}{dt} = 0$. We denote $X_i^*$ the random variable to which $X_i(t)$ converges in distribution $X_i^* \overset{d}{:=} \lim_{t \to \infty} X_i$. We denote $P^{\mathbb{M}}$ the equilibrium distribution of $\mathbf{X}^*$, and $P_{X_i^*}^{\mathbb{M}}$ the marginal probability of $X_i^*$.

**Equilibrium distribution of a Markov process model as a generative model** In the equilibrium distribution the place invariants in a Markov process model factorizes into a set of conditional

probability distributions, with a causal ordering based on the solutions to the master equations (see Supplementary materials for details). Based on this, the equilibrium distribution can be cast as a causal generative model $\mathbb{G}$ that consists of [20, 23]:

1. Random variables $\mathbf{X} = \{X_i; i \in 1...J\}$: the states of the system

2. A directed acyclic graph $\mathbb{D}$ with nodes $\{i \in J\}$ that impose an ordering on $\mathbf{X}$.

3. A set of probabilistic generative functions for each variable $X_i$, $\mathbf{p} = \{p_i, i \in J\}$ such that $X_i \sim p_i(\mathbf{PA}_{\mathbb{D},i}, N_i), \forall i \in J$ where $\mathbf{PA}_{\mathbb{D},i} \subseteq \mathbf{X} \setminus X_i$ are the parents of $X_i$ in $\mathbb{D}$.

$\mathbb{G}$ is a generative model that *entails* an observational distribution $P^{\mathbb{G}}$. This means that a procedure that first samples from each $\mathbf{p_i}$ along the ordering in $\mathbb{D}$ generates samples from $P^{\mathbb{G}}$. This is viewed as the generating process for the observed $\mathbf{X}$. A primary contribution of this work is a method for transforming $\mathbb{G}$ into and structural causal model.

**Structural causal models (SCMs)** A structural causal model $\mathbb{C}$ of the same system has the same causal directed graph $\mathbb{D}$, ordering the same random variables $\mathbf{X}$. The model consists of [20, 23]:

1. A distribution $P^{\mathbb{C}}_{\mathbf{N}}$ on independent *noise* random variables $\mathbf{N} = \{N_i; i \in J\}$

2. A set of functions $\mathbf{f} = \{f_i, i \in J\}$ called *structural assignments*, such that $X_i = f_i(\mathbf{PA}_{\mathbb{C},i}, N_i), \forall i \in J$ where $\mathbf{PA}_{\mathbb{C},i} \subseteq \mathbf{X} \setminus X_i$ are the parents of $X_i$ in $\mathbb{D}$.

$\mathbb{C}$ is a generative model that entails $P^{\mathbb{G}}$, the same observational distribution as $\mathbb{G}$. For consistency, we refer to this distribution as $P^{\mathbb{C}}$ when discussed in the context of $\mathbb{C}$. This means that a procedure that first samples noise values from $P^{\mathbb{C}}_{\mathbf{N}}$, and then sets the values of $\mathbf{X}$ deterministically with $\mathbf{f}$, generates samples from $P^{\mathbb{C}}$. This is viewed as the generating process for the observed $\mathbf{X}$.

**Interventions in Markov process models and in SCMs** An SCM $\mathbb{C}$ uses *ideal interventions*, which replace a random variable with a fixed point value. These are represented with Pearl's "do" notation $\mathrm{do}(X_i = x)$ [10, 22], denoted. The intervention that sets $X_i$ to $x$ replaces the structural assignment $X_i = f_i(\mathbf{PA}_{\mathbb{C},i}, N_i)$ with $X_i = x$. The *intervention distribution* $P^{\mathbb{C};\mathrm{do}(X_i=x)}$ is entailed by $\mathbb{C}$ under the intervention and is generally different from the equilibrium distribution $P^{\mathbb{M}}$ of $\mathbf{X}^*$.

In the context of a Markov process model, a typical intervention definition is that an intervention increases a reaction rate (catalyzation) or decreases a reaction rate (inhibition). We define a type of *soft intervention* [10] for Markov process models that make this rate manipulation comparable to the SCM's ideal intervention. We define a fixed post-equilibrium expected value for a variable that we want to achieve, then find a change to the variables rate parameter values that achieve that outcome. For example, an intervention that sets the equilibrium value of $X_i$ to $x$ does so by finding manipulating $X_i$'s rate parameters to achieve this result. Borrowing the "do" notation, denote this as $\mathrm{do}(X_i^* = x)$. Let the equilibrium distribution under intervention be $P^{\mathbb{M};\mathrm{do}(X_i^*=x)}$. We compare intervention queries on $P^{\mathbb{M};\mathrm{do}(X_i^*=x)}$ to $P^{\mathbb{C};\mathrm{do}(X_i=x)}$. For both Markov process models and SCMs, the intervention queries are answered by sampling from these distributions. See Supplementary materials for contrasts to related intervention modeling approaches.

**Counterfactual inference in SCMs** Counterfactual inference is the process of observing a random outcome, making inference about the unseen causes of the outcome, and then inferring the outcome that would have been observed under an intervention [23, 26]. For example, an SCM $\mathbb{C}$ helps answer the query "*Having observed $X_i = x$, what would have happened under the intervention $do(X_i = \neg x)$?*". SCMs support the following algorithm for counterfactual inference [2]: (1) having observed $X = x$, infer the noise distribution conditional on the observation $P^{\mathbb{C};X=x}_{\mathbf{N}}$, (2) replace $P^{\mathbb{C}}_{\mathbf{N}}$ with $P^{\mathbb{C};X=x}_{\mathbf{N}}$ in $\mathbb{C}$, (3) apply the intervention $\mathrm{do}(X = \neg x)$, and (4) sample from the resulting mutated model. The intuition is that in (2) we infer the latent initial conditions (values of $N$) that could have lead to the outcome $X = x$, this information is encoded in $P^{\mathbb{C};X=x}_{\mathbf{N}}$, the posterior of $N$ given $X = x$. We then pass that encoded information to the counterfactual world where $X$ is set to $\neg x$ and play out scenarios in that world by sampling from $P^{\mathbb{C};X=x}_{\mathbf{N}}$ and deriving downstream variables given those noise values. Thus the algorithm mutates $\mathbb{C}$ into an SCM entailing the *counterfactual distribution* $P^{\mathbb{C};X=x,\mathrm{do}(X=\neg x)}$.

## 3 Methods

### 3.1 Motivating example

This manuscript contributes a practical framework for casting Markov process models of a system observed at equilibrium as an SCM, for the purposes of conducting counterfactual inference. As a motivating example, we consider a system of three biomolecules (i.e., components) $X_1$, $X_2$ and Y. Each component takes two states: active ("on") and inactive ("off"). Component $X_1$ in the "on" state activates Y; component $X_2$ in the "on" state deactivates Y, as shown in the causal diagram [1] below:

$$X_1^{\text{on}} + Y^{\text{off}} \xrightarrow{v_1} X_1^{\text{on}} + Y^{\text{on}} \text{ and } X_2^{\text{on}} + Y^{\text{on}} \xrightarrow{v_2} X_2^{\text{on}} + Y^{\text{off}} \tag{2}$$

Let $X_1(t)$, $X_2(t)$, and $Y(t)$ be the total number of active-state particles of $X_1$, $X_2$, and Y at time $t$. Assume that each component has $T = 100$ particles in total, such that $T - Y(t)$ is the number of inactive particles of Y at time $t$, and that each component is initialized with 100 off-state particles.

To ensure that the equilibrium distribution of the Markov process model $\mathbb{M}$ has a closed-form solution, we limit this work to $\mathbb{M}$ with zero or first-order hazard functions (i.e. hazard functions for which outputs are either constant or directly proportional to a product of the inputs) [15, 29]. In this example, the hazard functions assume *mass action kinetics* [13], a common assumption in biochemical modeling. Let $h_1(Y(t))$ and $h_2(Y(t))$ denote stochastic rate laws for the activation and deactivation of Y, expressing the probabilities that the reactions occur in the instant $(t, t + dt]$. Then, according to a first-order stochastic kinetic assumption of chemical reactions [28], $h_1$ and $h_2$ are

$$h_1(Y(t)) = v_1 X_1(t)(T - Y(t)) \text{ and } h_2(Y(t)) = v_2 X_2(t)Y(t) \tag{3}$$

The hazard functions are parameterized by $\mathbf{v} = \{v_1, v_2\}$ regulating $X_1$ and $X_2$, and by the initial states.

The Kolmogorov forward equations determine the change in $P_{Y(t)}^{\mathbb{M}}$ as the system evolves in time:

$$\frac{dP_{Y(t)}^{\mathbb{M}}}{dt} = \left( h_1(Y(t) - 1)P_{Y(t)-1}^{\mathbb{M}} - h_1(Y(t))P_{Y(t)}^{\mathbb{M}} \right) + \left( h_2(Y(t) + 1)P_{Y(t)+1}^{\mathbb{M}} - h_2(Y(t))P_{Y(t)}^{\mathbb{M}} \right) \tag{4}$$

We pose a counterfactual query "*Having observed $X_1 = 34, X_2 = 45, Y = 56$, what would Y have been if $X_1$ was set to 50*"?

### 3.2 Converting a Markov process model into an SCM

Algorithm 1 summarizes the proposed steps of converting the Markov process model into an SCM. The steps are a series of mathematical derivations (as opposed to a pseudocode for a computational implementation). Below we illustrate these steps for the component Y in the motivating example. Additional mathematical details are available in Supplementary materials.

---

**Algorithm 1** *Convert Markov process into SCM*

Inputs:    Markov process model $\mathbb{M}$
Output:    Structural causal model $\mathbb{C}$

---
1: **procedure** GETSCM($\mathbb{M}$)
2:    ▶ *Solve master equation*
3:    $P^{\mathbb{M}}(t) := \int_t \frac{dP^{\mathbb{M}}(t)}{dt})dt$
4:    ▶ *Find the equilibrium distribution*
5:    $P^{\mathbb{M}} = \lim_{t \to \infty} P^{\mathbb{M}}(t)$
6:    ▶ *Use $P^{\mathbb{M}}$ to define generative model $\mathbb{G}$*
7:    $\mathbb{G} := \{x \sim P^{\mathbb{M}}\}$
8:    ▶ *Convert the generative model to an SCM*
9:    ▶ *that entails $P^{\mathbb{M}}$*
10:    $\mathbb{C} := \begin{cases} \mathbf{N} \sim P_{\mathbf{N}}^{\mathbb{C}} \\ \mathbf{X} = \mathbf{f}(\mathbf{X}, \mathbf{N}) \end{cases} : P^{\mathbb{C}} \approx P^{\mathbb{M}}$
11:    **return** $\mathbb{C}$

---

**Algorithm 2** *Counterfactual inference on SCM*

Inputs: Prior distribution on exogenous noise NPrior
       Structural causal model $\mathbb{C}$
       Observed endogenous variables $X = x$
       Counterfactual interventions $X = \neg x$
       Desired sample size $ssize$
Output: $ssize$ samples from $P^{\mathbb{C}; X=x, do(X:=\neg x)}$

---
1: **procedure** CFQUERY($\mathbb{C}$, NPrior, $x$, $\neg x$, $ssize$)
2:    ▶ *Create "observation" and "intervention" models*
3:    **obsModel** $\leftarrow$ Condition($\mathbb{C}$, $X = x$)
4:    **intModel** $\leftarrow$ Do($\mathbb{C}$, $X = \neg x$)
5:    ▶ *Infer noise distribution with observation model*
6:    NPosterior $\leftarrow$ Infer(**obsModel**, NPrior)
7:    ▶ *Simulate from intervention model w/ updated noise*
8:    samples = array($ssize$)
9:    **for** i in (0:$ssize$) **do**
10:        samples[i] $\leftarrow$ **intModel**(NPosterior)
11:    **return** samples

---

**Solve the master equation** (Algo. 1 line 3). We can arrive at the solution for $P_{Y(t)}^{\mathbb{M}}$ in Eq. (4) indirectly by solving the ordinary differential equation on the expectation of $Y(t)$ over $P_{Y(t)}^{\mathbb{M}}$:

$$\frac{d}{dt}E(Y(t)) = v_1 X_1(t)T - (v_1 X_1(t) + v_2 X_2(t)) E(Y(t)) \tag{5}$$

This has an analytical solution, where:

$$\frac{E(Y(t))}{T} = e^{-t(v_1 X_1(t) + v_2 X_2(t))} + \frac{v_1 X_1(t)}{v_1 X_1(t) + v_2 X_2(t)} \tag{6}$$

Finally, $Y(t)$ is a count of binary state variables with the same probability of being activated at a given instant. Then $P_{Y(t)}^{\mathbb{M}}$ must be Binomial distribution with $T$ trials, and trial probability $\frac{E(Y(t))}{T}$.

**Find the equilibrium distribution** (Algo. 1 line 5). Taking the limit in time of Eq. (6):

$$\frac{E(Y)}{T} = \lim_{t\to\infty} \frac{E(Y(t))}{T} = \frac{v_1 X_1(t)}{v_1 X_1(t) + v_2 X_2(t)} \tag{7}$$

Thus at equilibrium $Y$ follows the Binomial probability distribution with parameter $\frac{v_1 X_1(t)}{v_1 X_1(t) + v_2 X_2(t)}$.

**Use $P^{\mathbb{M}}$ to define generative model** $\mathbb{G}$ (Algo. 1 line 7). Let $\theta_{X_1}$ and $\theta_{X_1}$ be the probability parameters for the equilibrium Binomial distributions $P_{X_1}^{\mathbb{M}}$ and $P_{X_2}^{\mathbb{M}}$. Let $\theta_Y(X_1, X_2) = \frac{E(Y)}{T}$ be the probability parameter for the equilibrium Binomial distribution $P_Y^{\mathbb{M}}$. Define a generative model $\mathbb{G}$:

$$\mathbb{G} := \{X_1 \sim \text{Binom}(T, \theta_{X_1}); \ X_2 \sim \text{Binom}(T, \theta_{X_2}); \ Y \sim \text{Binom}(T, \theta_Y(X_1, X_2))\} \tag{8}$$

**Convert the generative model to an SCM that entails $P^{\mathbb{M}}$** (Algo. 1 line 10). We rely on a method of *monotonic conversion*, which restricts the class of possible SCMs to those with a common set of identifiable counterfactual quantities (such as the *probability of necessity*, i.e. the probability that Y would not have been activated without $X_1$) [20]. For each structural assignment $X_i = f_i(\mathbf{PA}_{\mathbb{C},i}, N_i), \forall i \in J$ the method enforces the property $E[X_i \mid do(\mathbf{PA}_{\mathbb{C},i} = y)] \geq E[X_i \mid do(\mathbf{PA}_{\mathbb{C},i} = y')] \Rightarrow f_i(y, n_i) \geq f_i(y', n_i) \forall n_i$.

For this example we selected a monotonic conversion by means of the inverse CDF transform. Denote $F^{-1}(u, n, p)$ the inverse CDF of the Binomial distribution, where $0 < u < 1$, and $n$ (number of trials) and $p$ (success probability) are the parameters of the Binomial distribution. Then the SCM $\mathbb{C}$ that entails $P^{\mathbb{M}}$ is defined as

$$N_{X_1}, N_{X_2}, N_Y \overset{\text{ind}}{\sim} \text{Uniform}(0, 1); \tag{9}$$

$$\mathbb{C} := \{X_1 = F^{-1}(N_{X_1}, T, \theta_{X_1}); \ X_2 = F^{-1}(N_{X_2}, T, \theta_{X_2}); \ Y = F^{-1}(N_Y, T, \theta_Y(X_1, X_2))\}$$

For larger models such as in Case studies 1 and 2 thereafter, it may be desirable to work with alternative transforms that are more amenable to gradient-based inference such as stochastic variational inference.

### 3.3 Counterfactual inference and evaluation

Algorithm 2 details the counterfactual inference on $\mathbb{C}$. Algorithms 3 and 4 in Supplementary materials detail the evaluation. The evaluation stems from the insight that noise at the equilibrium captures the stochasticity in the Markov process trajectories. Therefore, we repeatedly simulate pairs of the trajectories with and without the counterfactual intervention, with a same random seed in a pair, such that each pair has an identical stochastic component. We then compare the differences in the values of these pairs at equilibrium to the differences between the original and the intervened-upon values projected by the SCM. These differences estimate the respective *causal effects*. The algorithms differ in choosing a deterministic or a stochastic approach for the estimation of causal effects. To ensure scalability to large models and the ability to do inference over a broad set of structural assignments, we implemented the algorithms in PyTorch and the probabilistic programming language *Pyro* [4]. The code and the runtime data are in Supplementary materials.

## 4 Case studies

### 4.1 Case Study 1: The MAPK signaling pathway

**The system** The mitogen-activated protein kinase (MAPK) pathway is important in many biological processes, such as determination of cell fate. It is a cascade of three proteins, a MAPK, a MAPK

| | MAP3K | MAP2K | MAPK |
|---|---|---|---|
| activation hazard | $v_{K3}^{act}E1(T_{K3} - K3(t))$ | $v_{K2}^{act}K3(T_{K2} - K2(t))$ | $v_{K}^{act}K2(T_{K} - K(t))$ |
| deactivation hazard | $v_{K3}^{inh}K3(t)$ | $v_{K2}^{inh}K2(t)$ | $v_{K}^{inh}K(t)$ |

Table 1: The hazard functions in Case study 1 (MAPK), specified according to mass action enzyme kinetics.

kinase (MAP2K), and a MAPK kinase kinase (MAP3K), represented with a causal diagram [14, 24]

$$E1 \rightarrow MAP3K \rightarrow MAP2K \rightarrow MAPK \tag{10}$$

Here E1 is an input signal to the pathway. The cascade relays the signal from one protein to the next by changing the count of proteins in an active state.

**The biochemical reactions** A protein molecule is in an active state if it has one or more attached phosphoryl groups. Each arrow in Eq. (10) combines the reactions of phosphorylation (i.e., activation) and dephosphorylation (i.e., desactivation). For example, $E1 \rightarrow MAP3K$ combines two reactions

$$E1 + MAP3K \overset{v_{K3}^{act}}{\Rightarrow} E1 + P\text{-}MAP3K \text{ and } P\text{-}MAP3K \overset{v_{K3}^{inh}}{\Rightarrow} MAP3K \tag{11}$$

In the first reaction in Eq. (11), a particle of the input signal E1 binds (i.e., activates) a molecule of MAP3K to produce MAP3K with an attached phosphoryl. The rate parameter associated with this reaction is $v^{act}$. In the second reaction, phosphorylated MAP3K loses its phosphoryl (i.e., deactivates), with the rate $v^{inh}$. The remaining arrows in Eq. (10) aggregate similar reactions and rate pairs.

**The mechanistic model** Let $K3(t)$, $K2(t)$ and $K(t)$ denote the counts of phosphorylated MAP3K, MAP2K, and MAPK at time $t$. Let $T_{K3}$, $T_{K2}$, and $T_{K}$ represent the total amount of each of the three proteins, and $E1$ the total amount of input, which we assume are constant in time. We model the system as a continuous-time discrete-state Markov process $\mathbb{M}$ with hazard rates functions in Table 1.

**The data** We simulated the counts of protein particles using the Markov process model with rate parameters $v_{K3}^{act}$, $v_{K2}^{act}$, $v_{K}^{act}$ and deactivation rate parameters $v_{K3}^{inh}$, $v_{K2}^{inh}$, $v_{K}^{inh}$. We conducted three simulation experiments with three sets of rates, all consistent with a low concentration in a cell-sized volume (see Supplementary materials). The initial conditions assumed 1 particle of E1, 100 particles of the unphosphorylated form of each protein, and 0 particles of the phosphorylated form.

**The counterfactual of interest** Let K3, K2 and K denote the observed counts of phosphorylated MAP3K, MAP2K, and MAPK at 100 seconds, the time corresponding to an equilibrium for all the rates. Let K3$'$ be the count of phosphorylated MAP3K generated by a 3 times smaller $v_{K3}^{act}$. Thus $\mathbf{v}' = [v_{K3}^{act}/3, v_{K3}^{inh}, v_{K2}^{act}, v_{K2}^{inh}, v_{K}^{act}, v_{K}^{inh}]$. We pose the counterfactual question: *"Having observed the equilibrium particle counts K3, K2 and K, what would have been the count of K if we had K3$'$?"*.

**The evaluation** We derive the SCM $\mathbb{C}$ of the Markov process model and evaluate the counterfactual distribution $P_{K3}^{\mathbb{C};K3=\mathbf{x},K2=\mathbf{y},K=\mathbf{z},do(K3=x')}$ where $x'$ is the expected equilibrium value associated with $v'$. We evaluate this counterfactual statement as described in Algorithms 3 and 4 (with 500 seeds). If the counterfactuals from the converted SCMs are consistent with the Markov process models, their histograms from Algorithms 3 and 4 should overlap.

**The evaluation under model misspecification** We consider the Markov process model $\mathbb{M}$ with the first set of rates (see Supplementary materials). Let $[\mathbf{x}, \mathbf{y}, \mathbf{z}]$ be sampled from $\mathbb{M}$. Next, instead of the correct model we consider a misspecified model $\mathbb{M}'$, where $v_{K2}^{act}$ is perturbed with noise sampled from Uniform$(0.1, 0.5)$. We denote as $\mathbb{C}'$ the SCM corresponding to $\mathbb{M}'$, and evaluate the counterfactual distribution $P_{K3}^{\mathbb{C}';K3=\mathbf{x},K2=\mathbf{y},K=\mathbf{z},do(K3=x')}$. We expect that, since the counterfactual distribution from $\mathbb{C}'$ incorporates the data from the correct model, it should be closer to the true causal effect simulated from $\mathbb{M}$ than the direct simulation from the misspecified $\mathbb{M}'$. We repeat this experiment 50 times.

## 4.2 Case Study 2: The IGF signaling system

**The system** The growth factor signaling system is involved in growth and development of tissues. When external stimuli activate the epidermal growth factor (EGF) or the insulin-like growth factor (IGF), this triggers a cascade [3] in Fig. (1)(a). The Raf-Mek-Erk pathway is equivalent to Eq. (10), renamed to follow the convention adopted by the biological literature in this context.

**The biochemical reactions** All the edges in Fig. (1)(a) represent enzyme reactions $E + S \overset{v}{\rightarrow} E + P$, where the change of substrate S to product P is catalyzed by enzyme E. As in Case study 1, the

pointed edges combine activation and deactivation. The flat-headed edges only represent deactivation. The mechanistic model is built similarly to Case study 1.

**The data** We simulated the counts of protein particles using the Markov process model with rates in Supplemental Tables 2 and 3. The other settings are as in Case study 1. The initial condition assumed 37 particles of EGFR, 5 particles of IGFR, 100 particles of the unphosphorylated form of other proteins, and 0 particles of the phosphorylated form.

**The counterfactual of interest** Let $R'$ be the number of phosphorylated particles of Ras at equilibrium, achieved with $v'^{\text{act}}_{\text{Ras-SOS}} = v^{\text{act}}_{\text{Ras-SOS}}/6$. We pose the counterfactual: *Having observed the number of phosphorylated particles of each protein before the intervention, what would be the number of particles of Erk if the intervention had fixed Ras = $R'$?* Unlike the MAPK pathway, where the intervention on MAP3K affects the counterfactual target MAPK through a direct path, this system has two paths from Ras to Erk. One path goes directly through Raf, and the other through a mediating path PI3K $\rightarrow$ AKT. This challenges the algorithm to address multiple paths of influence.

**The evaluation** We consider the rates $v^{\text{act}}_{\text{Ras-SOS}}/6$, the counterfactual distribution $P^{\mathbb{C};X_i=x_i,\text{do(Ras}=R')}_{\text{Erk}}$, and the Algorithms 3 and 4 (with 300 seeds).

**The evaluation under model misspecification** We consider Markov process model $\mathbb{M}$ with the same rates and initial conditions as above. Let $x_i$ be sampled from $\mathbb{M}$. We then introduce a misspecified model $\mathbb{M}'$, where $v^{act}_{\text{AKT-PI3K}}$ is perturbed with noise sampled from Uniform$(0.01, 0.1)$. We denote as $\mathbb{C}'$ the SCM corresponding to $\mathbb{M}'$, and evaluate the counterfactual distribution $P^{\mathbb{C}';X_i=x_i,\text{do(Ras}=R')}_{\text{Erk}}$. The resulting counterfactual distribution from $\mathbb{C}'$ should be closer to the true causal effect simulated from $\mathbb{M}$ than the direct simulation from the misspecified $\mathbb{M}'$. We repeat this experiment 50 times.

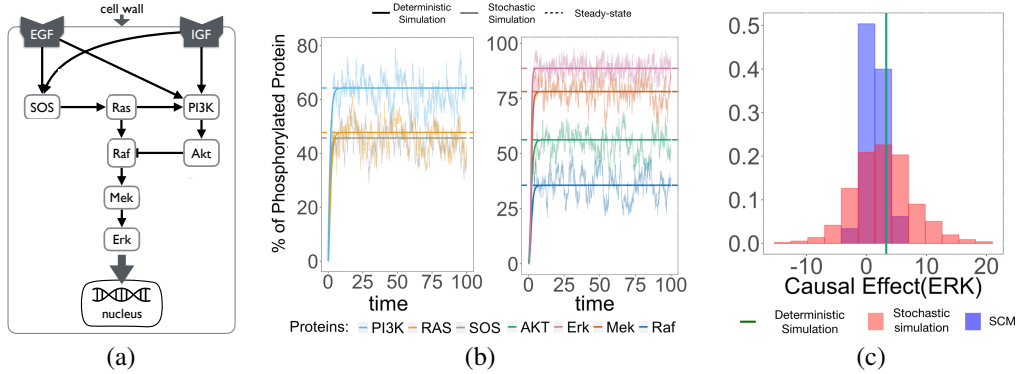

(a)        (b)        (c)

Figure 1: **Case study 2 (IGF).** (a) IGF signaling. The top nodes are receptors for the epidermal growth factor (EGF) and the insulin growth factor (IGF). Downstream of the receptors are several canonical signaling pathways (including Raf-Mek-Erk, a renamed equivalent of Eq. (10)). Each reaction has a single rate parameter. The auto-deactivation reactions are not pictured. (b) Deterministic and stochastic trajectories of the active-state proteins in the system. Horizontal lines are the expected values at equilibrium. (c) Histogram of causal effects, defined as differences between the "observed" and the "counterfactual" trajectories of ERK at equilibrium.

## 5 Results

### 5.1 Case Study 1: The MAPK signaling pathway

**Solve stochastic process's master equation** (Algorithm 1 line 3). As in the motivating example, we indirectly solve $\frac{dP^{\mathbb{M}}(t)}{dt}$ by way of the solving the forward equations for the expectation. For K3$(t)$ this is $\frac{dE(\text{K3}(t))}{dt} = v^{act}_{\text{K3}}\text{E1}\left(T_{\text{K3}} - E(\text{K3}(t)) - v^{inh}_{\text{K3}}E(\text{K3}(t))\right)$ (Added Expectation in RHS, please review). We derive similar forward equations for K2$(t)$ and K$(t)$. We solve the ODE above:

$$\frac{E(\text{K3}(t))}{T_{\text{K3}}} = e^{-t(v^{act}_{\text{K3}}\text{E1}+v^{inh}_{\text{K3}})} + \frac{v^{act}_{\text{K3}}\text{E1}}{v^{act}_{\text{K3}}\text{E1} + v^{inh}_{\text{K3}}} \tag{12}$$

and obtain the equilibrium by taking the limit $t \rightarrow \infty$. The first term in Eq. (12) goes to 0:

$$\frac{E(\text{K3})}{T_{\text{K3}}} = \frac{v^{act}_{\text{K3}}T_{\text{K3}}\text{E1}}{v^{act}_{\text{K3}}\text{E1} + v^{inh}_{\text{K3}}} \tag{13}$$

**Find the equilibrium distribution** (Algorithm 1 line 5) As in Sec. 3.2), the each active-state MAPK protein has a Binomial marginal distribution. Let $\theta_{K3}(E1)$ denote the probability that a MAP3K particle is active at equilibrium given E1. After solving the master equation,

$$\theta_{K3}(E1) = \frac{E(K3)}{T_{K3}} = \frac{v_{K3}^{act}E1}{v_{K3}^{act}E1 + v_{K3}^{inh}} \tag{14}$$

Extending this solution to MAP2K and MAPK leads to probabilities

$$\theta_{K3}(E1) = \frac{v_{K3}^{act}E1}{v_{K3}^{act}E1 + v_{K3}^{inh}}; \ \theta_{K2}(K3) = \frac{v_{K2}^{act}K3}{v_{K2}^{act}K3 + v_{K2}^{inh}}; \ \theta_{K}(K2) = \frac{v_{K}^{act}K2}{v_{K}^{act}K2 + v_{K}^{inh}} \tag{15}$$

and the following equilibrium distributions:

$$P_{K3}^{\mathbb{M}} \equiv \mathrm{Binomial}(T_{K3}, \theta_{K3}(E1)); \ P_{K2}^{\mathbb{M}} \equiv \mathrm{Binomial}(T_{K2}, \theta_{K2}(K3)); \ P_{K}^{\mathbb{M}} \equiv \mathrm{Binomial}(T_{K}, \theta_{K}(K2)) \tag{16}$$

**Use $P^{\mathbb{M}}$ to define generative model $\mathbb{G}$** (Algorithm 1 line 7). From here it is straightforward to create a generative model that entails $P_{K3}^{\mathbb{M}}$:

$$\mathbb{G} := \{K3 \sim \mathrm{Binom}(T_{K3}, \theta_{K3}(E1)); \ K2 \sim \mathrm{Binom}(T_{K2}, \theta_{K2}(K3)); \ K \sim \mathrm{Binom}(T_{K}, \theta_{K}(K2))\} \tag{17}$$

**Convert the generative model to an SCM that entails $P^{\mathbb{M}}$** (Algorithm 1 line 10). Here the challenge is in expressing the stochasticity in $\mathbb{G}$, while defining K3, K2, K as deterministic functions of the noise variables $N_K, N_{K2}, N_{K3}$. Instead of using the inverse binomial CDF, we demonstrate the use of a differentiable monotonic conversion, so that we can validate approximate counterfactual inference with stochastic gradient descent. We achieve this by first applying a Gaussian approximation to the Binomial distribution, and then applying the "reparameterization trick" used in variational autoencoders [25] (combined in helper function $q$ in Eq. (18)).

$$N_K, \ N_{K2}, N_{K3} \stackrel{\mathrm{ind}}{\sim} N(0, 1); \ q(\theta, T, N) = N \cdot (T\theta(1-\theta))^{1/2} + \theta T \tag{18}$$

$$\mathbb{C} := \{K3 = q(\theta_{K3}(E1), T_{K3}, N_{K3}); \ K2 = q(\theta_{K2}(K3), T_{K2}, N_{K2}); \ K = q(\theta_K(K2), T_K, N_K)\} \tag{19}$$

The Gaussian approximation facilitates the gradient-based inference in line 6 of Algorithm 2. Despite the approximation, the resulting SCM is still defined in terms of $\theta$. In this manner the SCM retains the biological mechanisms and the interpretation of the Markov process model.

**Create "observation" and "intervention" models** (Algorithm 2 lines 3-4) In a probabilistic programming language, the deterministic functions in Eq. (19) are specified with a Dirac Delta distribution. However, at the time of writing, gradient-based inference in *Pyro* produced errors when conditioning on a Dirac sample. We relaxed the Dirac Delta to allow a small amount of density.

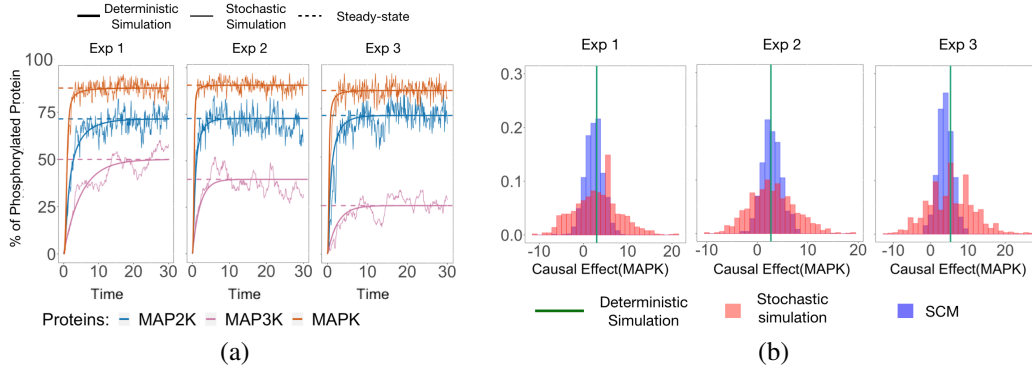

Figure 2: **Case study 1 (MAPK)**. (a) Deterministic and stochastic trajectories of the active-state MAPK proteins. Horizontal lines are the expected values at equilibrium. (b) Histograms of causal effects, defined as differences between the "observed" and the "counterfactual" trajectories of MAP3K at equilibrium.

**Infer noise distribution with observation model** (Algorithm 2 line 6) We use stochastic variational inference ([12]) to infer and update $N_{K3}$, $N_{K2}$ and $N_K$ from the observation model, and independent Normal distributions as approximating distributions.

**Simulate from intervention model with updated noise** (Algorithm 2 line 10) After updating the noise distributions, we generate the target distribution of the intervention model.

**Deterministic and stochastic counterfactual simulation and evaluation** (Algorithms 3 and 4 in Supplementary materials). Fig. (2)(a) illustrates that the simulated trajectories converge in steady state.

Since we rely on the Gaussian approximation to the Binomial in constructing $\mathbb{C}$, we would expect worse results if we were to set the rates on or near the boundaries 0 and 100, where the approximation is weak. Fig. (2)(b) shows that for each experiment with different sets of rates, the causal effects from the SCM's counterfactual distribution are centered around the ground truth simulated deterministically using Eq. (12) and similar equations for K2 and K. The SCM's distribution has less variance, likely due to the fact that ideal interventions in the SCM allow less variation than rate-manipulation-based interventions in the Markov process model.

**Evaluation under model misspecification** Fig. (3)(a) shows histograms from one of the 50 repetitions of the experiment conducted to evaluate the robustness of the SCM under model misspecification, and illustrates that the causal effects from the misspecified SCM is closer to true causal effect than the causal effect derived from a direct but misspecified simulation. Over the 50 repetitions, the absolute difference between the median of the true causal effect and the causal effect derived from the misspecified SCM is on average 0.343. The absolute difference between the median of the true causal effect and misspecified direct simulation is on average 1.03.

## 5.2 Case Study 2: The IGF signaling system

The derivations for the growth factor signaling system align closely with that of the motivating example and of the MAPK model. For variable $X_i$ with parents $\mathbf{PA}_{\mathbb{M},i}$, we partition each parent set into activators and inhibitors $\mathbf{PA}_{\mathbb{M},i} = \{\mathbf{PA}_{\mathbb{M},i}^{\mathrm{act}}, \mathbf{PA}_{\mathbb{M},i}^{\mathrm{inh}}\}$. The rate parameters are also partitioned into $\mathbf{v} = \{\mathbf{v}^{\mathrm{act}}, \mathbf{v}^{\mathrm{inh}}\}$. For each $X_i$ the probability for particle activation at equilibrium is:

$$\theta_{X_i}(\mathbf{PA}_{\mathbb{M},i}) = \frac{\mathbf{v}^{\mathrm{act}}\mathbf{PA}_{\mathbb{M},i}^{\mathrm{act}}}{\mathbf{v}^{\mathrm{act}}\mathbf{PA}_{\mathbb{M},i}^{\mathrm{act}} + \mathbf{v}^{\mathrm{inh}}\mathbf{PA}_{\mathbb{M},i}^{\mathrm{inh}}} \tag{20}$$

Next, we derive an SCM using the same Normal approximation to the Binomial distribution as in the MAPK pathway. Fig. (1)(b) plots deterministic and stochastic time courses for active states counts of the proteins in the pathway. Fig. (1)(c) illustrates that the counterfactual inference was successful despite the increased model complexity and size.

**The evaluation under model misspecification** Similarly to Case study 1, Fig. (3)(b) illustrates that the causal effects from the misspecified SCM are closer to the true causal effect than the causal effect derived from a direct but misspecified simulation. Over the 50 repetitions, the absolute difference between the median of the true causal effect and the causal effect derived from the misspecified SCM is on average 7.563. The absolute difference between the median of the true causal effect and misspecified direct simulation is on average 92.55.

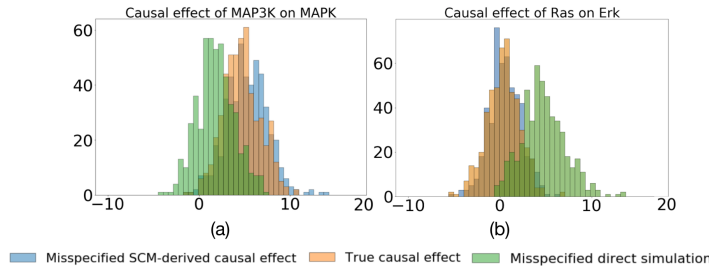

Figure 3: Histograms of causal effects, i.e. differences between the "observed" and the "counterfactual" trajectories at equilibrium, for one repetition of the evaluation. The causal effect from the misspecified SCM (blue histogram) is closer to true causal effect (orange histogram) than the causal effect derived from a direct but misspecified simulation (green histogram). (a) MAPK, (b) IGF.

## 6 Discussion

This work proposed a practical approach for casting a Markov process model of a system at equilibrium as an SCM. Equilibrium counterfactual inferences using this SCM are anchored to the rate laws of the Markov process. We derived the specific steps of conducting counterfactual inference in real-life case studies of biochemical networks. The case studies illustrate that the counterfactual inference is consistent with the differences in the initial and the intervened upon trajectories of the Markov process, and makes the selection of interventions more robust to model misspecification. This approach opens many opportunities for future methodological research, such as extending this approach to models with cycles, a common feature of complex systems. Overall, this work is a step towards broader adoption of counterfactual inference in systems biology and other applications.

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
