[Supplementary Material]

# Supplementary materials: Integrating Markov processes with structural causal modeling enables counterfactual inference in complex systems

**Robert Ness**
Gamalon Inc.
robert.ness@gamalon.com

**Kaushal Paneri**
Northeastern University
kaushalpaneri@gmail.com

**Olga Vitek**
Northeastern University
o.vitek@northeastern.edu

## 1  Supplementary data and algorithms

| | $v_{\text{K3}}^{\text{act}}$: MAP3K on | $v_{\text{K3}}^{\text{inh}}$: MAP3K off | $v_{\text{K2}}^{\text{act}}$: MAP2K on | $v_{\text{K2}}^{\text{inh}}$: MAP2K off | $v_{\text{K}}^{\text{act}}$: MAPK on | $v_{\text{K}}^{\text{inh}}$: MAPK off |
|---|---|---|---|---|---|---|
| Exp 1 | 0.1 | 0.1 | 0.1 | 2.0 | 0.1 | 1.0 |
| Exp 2 | 0.2 | 0.3 | 0.2 | 3.0 | 0.2 | 1.5 |
| Exp 3 | 0.1 | 0.3 | 0.5 | 5.0 | 0.3 | 4.0 |

Table 1: The rates parameters in Case study 1 (MAPK). Each row corresponds to a set $\mathbf{v}$.

| | $v_{\text{SOS-EGFR}}^{\text{act}}$ | $v_{\text{SOS-IGFR}}^{\text{act}}$ | $v_{\text{Ras-SOS}}^{\text{act}}$ | $v_{\text{PI3K-EGFR}}^{\text{act}}$ | $v_{\text{PI3K-IGFR}}^{\text{act}}$ | $v_{\text{PI3K-Ras}}^{\text{act}}$ | $v_{\text{AKT-PI3K}}^{\text{act}}$ | $v_{\text{Raf-Ras}}^{\text{act}}$ | $v_{\text{Raf-AKT}}^{\text{act}}$ | $v_{\text{Mek-Raf}}^{\text{act}}$ | $v_{\text{Erk-Mek}}^{\text{act}}$ |
|---|---|---|---|---|---|---|---|---|---|---|---|
| Rates | 0.01 | 0.01 | 0.01 | 0.01 | 0.01 | 0.01 | 0.01 | 0.01 | 0.01 | 0.05 | 0.05 |

Table 2: The activation rates parameters in Case study 2 (IGF). Activation rates are formatted as $v_{\text{child-parent}}^{\text{act}}$

| | $v_{\text{SOS}}^{\text{inh}}$ | $v_{\text{Ras}}^{\text{inh}}$ | $v_{\text{PI3K}}^{\text{inh}}$ | $v_{\text{AKT}}^{\text{inh}}$ | $v_{\text{Raf}}^{\text{inh}}$ | $v_{\text{Raf-AKT}}^{\text{inh}}$ | $v_{\text{Mek}}^{\text{inh}}$ | $v_{\text{Erk}}^{\text{inh}}$ |
|---|---|---|---|---|---|---|---|---|
| Rates | 0.5 | 0.5 | 0.5 | 0.5 | 0.3 | 0.01 | 0.5 | 0.5 |

Table 3: The deactivation rates parameters in Case study 2 (IGF). All but one rate are auto-deactivation. $v_{\text{RAF-ACT}}^{\text{inh}}$ is formatted as $v_{\text{child-parent}}^{\text{inh}}$

**Runtime** All the experiments were performed on a Macbook pro with intel 2.2 GHz core i7 and 16 GB RAM. The counterfactual inference (Algorithm 2) on the MAPK and growth factor models with one sample was computed in 4.89 seconds and 10.4 seconds with stochastic variational inference. For the MAPK model, Algorithm 3 with trajectories of length 30 took 0.1 seconds, and Algorithm 4 with 1000 seeds took 1800 seconds. For the IGF model, the Algorithm 3 with trajectories of length 30 took 0.9 and Algorithm 4 with 300 seeds took 1245 seconds.

---

**Algorithm 3** *Deterministic counterfactual simulation*
*and evaluation with Markov process model*

Inputs:    Markov process model $\mathbb{M}$
              Rate sets $\mathbf{v}$ and $\mathbf{v}'$
              Equilibrium time point $T$
              Index of counterfactual intervention target $i$
              Index of counterfactual query target $j$
              Noise prior for SCM *NPrior*
Output:   Histogram of causal effects

---

1: **procedure** CF-DETERMINSIM($\mathbb{M}$, $\mathbf{v}$, $\mathbf{v}'$, $T$, $i$, $j$, NPrior)
2:     ▶ *Simulate expected value at equilbrium (Eq. (9) below) using* $\mathbf{v}$
3:     $\mathbf{x} = E(\mathrm{sim}(\mathbb{M}, \mathbf{v})[T])$
4:     ▶ *Simulate expected value at equilibrium (Eq. (9) below) using* $\mathbf{v}'$
5:     $\mathbf{x}' = E(\mathrm{sim}(\mathbb{M}, \mathbf{v}')[T])$
6:     ▶ *Calculate causal effects*
7:     **for** index $k$ in array $\Delta$ **do**
8:         $\delta_{\text{true}} = x'_j - x_j$
9:         ▶ *Simulate CF value from SCM*
10:       $\mathbb{C} = \mathrm{GetSCM}(\mathbb{M})$
11:       $x^*_j \sim \mathrm{CFQuery}(\mathbb{C}, \mathrm{NPrior}, \mathbf{x}, x'_i)$
12:       ▶ *Calculate difference*
13:       $\Delta[k] = x^*_j - x_j$
14:     ▶ *Compare $\Delta$ to $\delta_{true}$*
15:     $\mathrm{histogram}(\Delta, \text{verticle-line} = \delta_{\text{true}})$

---

---

**Algorithm 4** *Stochastic counterfactual simulation*
*and evaluation with Markov process model*

Inputs:    Markov process model $\mathbb{M}$
              Rate sets $\mathbf{v}$ and $\mathbf{v}'$
              Equilibrium time point $T$
               Index of counterfactual intervention target $i$
              Index of counterfactual query target $j$
              List of random seeds $S$
              Noise prior for SCM *NPrior*
*Output:*   *Histogram of causal effects*

---

1: **procedure** CF-STOCHSIM($\mathbb{M}$, $\mathbf{v}$, $\mathbf{v}'$, $T$, $i$, $j$, $S$, NPrior)
2:     ▶ *Simulate a deterministic equilibrium using* $\mathbf{v}'$
3:     $\mathbf{x}^{d'} = \mathrm{sim}(\mathbb{M}, \mathbf{v}')[T]$
4:     **for** index $k$ in S, & collectors $\Delta_{\mathbb{M}}$, $\Delta_{\mathbb{C}}$ **do**
5:         ▶ *Simulate a stochastic equilbrium using* $\mathbf{v}$
6:         $\mathbf{x}^s = \mathrm{sim}(\mathbb{M}, \mathbf{v}, \mathrm{seed} = S[k])[T]$
7:         ▶ *Simulate a stochastic equilibrium using* $\mathbf{v}'$
8:         $\mathbf{x}^{s'} = \mathrm{sim}(\mathbb{M}, \mathbf{v}', \mathrm{seed} = S[k])[T]$
9:         ▶ *Calculate causal effects*
10:       $\Delta_{\mathbb{M}}[k] = x^{s'}_j - x^s_j$
11:       ▶ *Simulate CF value from SCM using* $\mathbf{x}^{d'}$
12:       $\mathbb{C} = \mathrm{GetSCM}(\mathbb{M})$
13:       $x^*_j \sim \mathrm{CFQuery}(\mathbb{C}, \mathrm{NPrior}, \mathbf{x}, x^{d'}_i)$
14:       ▶ *Calculate difference*
15:       $\Delta_{\mathbb{C}}[k] = x^*_j - x^s_j$
16:     ▶ *Compare $\Delta_{\mathbb{C}}$ to $\Delta_{\mathbb{M}}$*
17:     $\mathrm{overlayHistograms}(\Delta_{\mathbb{C}}, \Delta_{\mathbb{M}})$

---

# 2 Supplementary methods and proofs

## 2.1 Markov process equilibrium as causal Bayesian network

In many domains, Markov process models can have arbitrary levels of granularity in terms of the components in the system and their interactions. This is certainly true in systems biology, where Markov process models (called stochastic kinetic models in this context) vary in details of chemical reactions between molecular species. In order to generalize the proposed procedure, we assume that the random variables in the Markov process model are *place invariants*.

**Definition 2.1. Place invariant** A place invariant (also called p-invariant or s-invariant) is a set of model components with a constant sum across all the model states. A minimal place invariant cannot be decomposed into smaller place invariants [4].

We assume a Markov process model with a unique equilibrium distribution and no cycles (in most cases, the cycles can be collapsed into a larger p-invariant). Therefore, the equilibrium distribution is factorized according to a directed acyclic graph, given by solving each $\frac{dP(X(t))}{dt}$ for each variable $X$ in the model. We cast this distribution as a causal Bayesian network, with the conditional probability distributions given by the equilibrium distributions.

## 2.2 Comparison to other intervention approaches

In causal inference literature, a commonly used definition of intervention is the "ideal" intervention, which *directly* sets the value of the target variable, and cuts off the influence of the targets' direct parents. This definition is adapted to dynamic processes in [2, 3, 5], where an ideal intervention fixes the target variable at a specific value throughout the transient states of the system until the equilibrium is reached, and then blocks all influence on the intervened variable with no side effects. The prior work on ideal interventions largely focused on the equivalence between the equilibrium outcome of the intervention on the dynamic model, and the "do"-style ideal interventions on an equilibrium SCM. Reference [3] refers to this as the *equilibrium-manipulation commutability property*.

This manuscript uses a different approach. We define an intervention on a Markov process as a manipulation of the parameter rates, to *indirectly* achieve a desired equilibrium value of a target variable. Unless the manipulation sets the reaction rate to zero, the parent variables are still influencing the intervention target upon the intervention. This definition of the intervention is motivated by the biological application, where it is common to have directly manipulable rates (e.g., through a catalyst), and where the equilibrium values are typically an indirect result of the manipulation.

In contrast to the prior work on ideal interventions in dynamic processes, this manuscript aims to cast an equilibrium probability model $\mathbb{G}$ as an SCM, in order to make useful counterfactual inference on that model. We show that with zero or first order hazard functions, we can work directly with the solutions to a system of ordinary differential equations on the expectations of each variable in the system. These constitute what [5] calls a *labeled set of equilibrium equations*.

The distinction between manipulable and non-manipulable causes is treated in depth in the causal inference literature [8, 9]. Those interested in working with ideal interventions with our proposed approach could view the solutions to the expectation equations as a "labeled set of equilibrium equations" in [5]. Since the expectation equations are deterministic, this can help investigate whether the equilibrium-manipulation commutability conditions in Lemma 1 of [5] could apply to ideal interventions *in expectation* with our approach.

Those interested in working with ideal interventions with our proposed approach could view the solutions to the expectation equations as a "labeled set of equilibrium equations" in [5]. Since the expectation equations are deterministic, this can help investigate whether the equilibrium-manipulation commutability conditions in Lemma 1 of [5] could apply to ideal interventions *in expectation* with our approach.

## 2.3 Details of the motivating example

### 2.3.1 Summary

Assume that species Y is regulated by species $X_1$ and $X_2$. A particle of any of these three species is in either an active or inactive state. $X_1$ is an activator of Y, meaning that an interaction with active $X_1$ converts inactive Y into active Y. $X_1$ is an deactivator. An interaction with active $X_2$ converts active Y to inactive Y. We can represent the reactions under a mass action kinetic assumption as follows:

$$X_1^{\text{on}} + Y^{\text{off}} \quad \overset{v_1}{\to} \quad X_1^{\text{on}} + Y^{\text{on}} \quad \text{and} \quad X_2^{\text{on}} + Y^{\text{on}} \quad \overset{v_2}{\to} \quad X_2^{\text{on}} + Y^{\text{off}} \qquad (1)$$

$v_1$ and $v_2$ are the rate parameters for the two reactions. In this example, $X_1$ is p-invariant, as the sum of active ("on") and inactive ("off") particles is constant. The same holds for $X_2$ and Y.

## 2.4 Building a probability model of the system

Let $X_1(t)$, $X_2(t)$, and $Y(t)$ represent the total *active-state* particle count in a cell of $X_1$, $X_2$, and Y respectively at time $t$. Let $T_y$ represent the total particle count (active and inactive) in a cell of Y, such that $T_y - Y(t)$ is the number of inactive particles of Y at time $t$.

Let $\pi(y, t) = P(Y(t) = y \mid T_y, X_1(t), X_2(t))$ represent the conditional probability distribution of $Y(t)$. Each particle of Y is in active state with some probability, i.e. a Bernoulli trial. Therefore $Y(t)$ is the sum of Bernoulli random variables such that $\pi(y, t)$ is a Binomial distribution with $T_y$ trials.

Let $\theta_y(X_1(t), X_2(t))$ denote the probability that a particle of Y is in active state at time $t$. This probability is needed to fully specify the Binomial distribution. The following derivation demonstrates how $\theta_y(X_1(t), X_2(t))$ is a function of $X_1(t), X_2(t)$.

### 2.4.1 Finding the equilibrium probability distribution for $Y(t)$

The *hazard rate function* of a biochemical reaction is the probability that the reaction occurs in a given instant. It is determined by the particle counts at that instant, and by the rate parameters. Let $h_1(Y(t))$ and $h_2(Y(t))$ represent the hazard rate functions for activation and deactivation respectively:

$$h_1(Y(t)) = v_1 X_1(t)(T_y - Y(t)) \qquad (2)$$
$$h_2(Y(t)) = v_2 X_2(t) Y(t) \qquad (3)$$

Let $S_1$ and $S_2$ denote the change in particle count after reactions in Eq. (1).

$$S_1 = 1$$
$$S_2 = -1$$

The Kolmogorov forward equations determine the change in $\pi(y, t)$ as the system evolves in time.

$$\frac{d}{dt}\pi(y, t) = \sum_{i=1}^{2} \left( h_i(y - S_i)\pi(y - S_i, t) - h_i(y)\pi(y, t) \right)$$

Let $E_\pi(.)$ denote the conditional expectation function over $\pi(y, t)$, i.e. $E_\pi(f(Y(t))) \equiv \sum_{y=0}^{T_y} f(y)\pi(y, t)$.

*Lemma* 2.1. If the hazard functions are zero or first order, then solving for equilibrium of $\frac{d}{dt}E_\pi(Y(t))$ yields the equilibrium solution to the $\frac{d}{dt}\pi(y, t)$

*Proof.* By Kolmogorov's forward equation, the change in the expectation of $Y(t)$ is

$$
\begin{aligned}
\frac{d}{dt} E_\pi(Y(t)) &= \frac{d}{dt} \sum_{y=0}^{T_y} y\pi(y,t) = \sum_{y=0}^{T_y} y \frac{d}{dt}\pi(y,t) \\
&= \sum_{y=0}^{T_y} y \left[ \sum_{i=1}^{2} (h_i(y-S_i)\pi(y-S_i,t) - h_i(y)\pi(y,t)) \right] \\
&= \sum_{i=1}^{2} \left( \sum_{y=0}^{T_y} yh_i(y-S_i)\pi(y-S_i \mid T_y, X_1(t), X_2(t)) - \sum_{y=0}^{T_y} yh_i(y)\pi(y,t) \right) \\
&= \sum_{i=1}^{2} \left( \sum_{y=0}^{T_y} (y+S_i)h_i(y)\pi(y,t) - \sum_{y=0}^{T_y} yh_i(y)\pi(y,t) \right) \\
&= \sum_{i=1}^{2} \left( E_\pi((Y(t)+S_i)h_i(Y(t))) - E_\pi(h_i(Y(t))) \right) \\
&= \sum_{i=1}^{2} S_i E_\pi(h_i(Y(t)))
\end{aligned}
$$

If hazards are zero or first order, then the linearity property of the expectation operator allows for an analytical solution. Without loss of generality, we demonstrate this with the motivating example. Substituting in the hazard functions in Eq. (3):

$$
\begin{aligned}
\frac{d}{dt} E_\pi(Y(t)) &= \sum_{i=1}^{2} S_i E_\pi(h_i(Y(t))) \\
&= E_\pi v_1 X_1(t)(T_y - Y(t)) - E_\pi v_2 X_2(t) Y(t) \\
&= v_1 X_1(t) T_y - v_1 X_1(t) E_\pi(Y(t)) - v_2 X_2(t) E_\pi(Y(t)) \\
&= v_1 X_1(t) T_y - (v_1 X_1(t) + v_2 X_2(t)) E_\pi(Y(t))
\end{aligned}
\tag{4}
$$

Let $\theta_y(X_1(t), X_2(t))$ denote the probability that a particle of Y is in active state at time $t$. $\theta_y(X_1(t), X_2(t))$ is a function $X_1(t), X_2(t)$. This follows from the fact that $\pi(y,t)$ has a Binomial distribution, and therefore $\theta_y$ is determined by the expectation $E_\pi(Y(t))$.

$$
\theta_y(X_1(t), X_2(t)) \equiv \frac{E_\pi(Y(t))}{T_y}
\tag{5}
$$

Substituting Equation 5 into 4:

$$
\frac{d}{dt} E_\pi(Y(t)) = v_1 X_1(t) T_y - (v_1 X_1(t) + v_2 X_2(t)) T_y \theta_y(X_1(t), X_2(t))
$$

$$
\frac{d}{dt} \theta_y(X_1(t), X_2(t)) = v_1 X_1(t) - (v_1 X_1(t) + v_2 X_2(t)) \theta_y(X_1(t), X_2(t))
\tag{6}
$$

The analytical solution is:

$$
\theta_y(X_1(t), X_2(t)) = e^{-t(v_1 X_1(t) + v_2 X_2(t))} + \frac{v_1 X_1(t)}{v_1 X_1(t) + v_2 X_2(t)}
\tag{7}
$$

Therefore the probability distribution of $Y(t)$ is given by

$$
Y(t) \sim \text{Binomial}(T_y, \theta_y(X_1(t), X_2(t)))
\tag{8}
$$

$\theta_y(X_1(t), X_2(t))$ achieves *steady-state* when $\frac{d}{dt}\theta_y(X_1(t), X_2(t)) = 0$. The solution is found by setting the left-hand side in Eq. (6) to 0, or alternatively, taking the limit in time of Eq. (7)

$$\lim_{t\to\infty} \theta_y(X_1(t), X_2(t)) = \frac{v_1 X_1(t)}{v_1 X_1(t) + v_2 X_2(t)} \tag{9}$$

$\theta_y(X_1(t), X_2(t))$ is the only component of the Binomial probability distribution of $Y(t)$ that varies in time. The steady-state solution of Eq. (7) also provides the steady-state distribution of $Y(t)$ (also referred to as stationary or invariant distribution in stochastic process literature). For simplicity we assume that the counts of active $X_1$ and $X_2$ are also the results of processes with stationary distributions. Let $Y$, $X_1$ and $X_2$ represent steady state active particle counts for Y, $X_1$, and $X_2$. Then the steady-state distribution of $Y$ is given by:

$$\theta_y(X_1, X_2) = \frac{v_1 X_1}{v_1 X_1 + v_2 X_2} \tag{10}$$

$$Y \sim \text{Binomial}(T_y, \theta_y(X_1, X_2)) \tag{11}$$

$\square$

## 2.5 Connections to causal constraint models

The rate laws in a dynamic model describe the mechanistic relationships between variables of the model. Our goal is to build a causal model of the system at equilibrium that is faithful to these relationships.

However, particular sets of interventions could mutate the mechanisms underpinning these relationships. For example, the mechanisms in the dynamic model may give rise to a conservation law, i.e., that the sum of the values of a particular subset of variables is constant in time, though the values themselves may vary. In would be possible to specify interventions that violate this conservation law. The equilibrium SCM model assumes that the mechanisms connected to this conservation law are invariant and has no way of prohibiting a set of interventions that violate this assumption. The SCM's predictions of the equilibrium behavior of a set of interventions that violate this conservation law would be inconsistent with simulated interventions from the dynamic model.

Blom et al. address this problem by introducing causal constraint models (CCMs), an extension to SCM models of the equilibrium of dynamic models [1]. CCMs explicitly identify sets of interventions for which the functional relationships derived from the underlying dynamic model are invariant.

Our work illustrates the process of converting a dynamic model to an equilibrium SCM and demonstrates counterfactual inferences consistent with the dynamic model ground truth. We apply simplifying assumptions to our dynamic models designed to avoid the conflicts between SCM intervention and the dynamic model mechanism that Blom et al.'s work addresses. In the following subsections, we show that the examples we use in this work do not have any intervention constraints that would warrant the use of a CCM.

The case studies used in this work are special cases of the enzyme kinetic reactions that take the form:

$$E + S \underset{K_{\text{off}}}{\overset{K_{\text{on}}}{\rightleftharpoons}} [ES] \overset{K_2}{\rightarrow} E + P$$

$$P \overset{v_{\text{off}}}{\rightarrow} S$$

where $E$ is a enzyme, $S$ is a substrate, and $P$ is a product. Blom et al. in constrast use the following enzyme model as an example:

$$\varnothing \overset{K_0}{\rightarrow} S$$

$$E + S \underset{K_{\text{-1}}}{\overset{K_1}{\rightleftharpoons}} [ES] \overset{K_2}{\rightarrow} E + P$$

$$P \overset{K_3}{\rightarrow} \varnothing$$

Our enzyme model is a special case of the Blom et al. model where substrate does not appear from nothing nor substrate disappear into nothing, but rather product converts back to product.

Further, we simplify the model such that it collapses over the intermediate compound (simplifying from Michaelis–Menten kinetics to mass action kinetics):

$$E + S \xrightarrow{v_{\text{on}}} E + P$$
$$P \xrightarrow{v_{\text{off}}} S$$

Finally, when modeling the hazard rates in our Markov process model, we incorporate the fact that in our model there is a conservation law between product and substrate. Let $P(t)$ and $S(t)$ be the total amount of product and substrate at time $t$, and $T = P(t) + S(t)$ be the unchanging total. Instead of modeling the production rate law of $P$ as $v_{\text{on}}E(t)S(t)$, we eliminate the variable $S(t)$ from our model and use $v_{\text{on}}E(t)(T - P(t))$. This, combined with the elimination of the compound $[ES]$ from the model, remove the need for the causal constraints outlined in Blom et al. enzyme model.

We believe that our work and Blom et al.'s CCM framework are complimentary. CCM's could be used to avoid making simplifying assumptions when they are not appropriate relative to the complexity of the dynamic system. Our approach of modeling the dynamic model as a Markov process in order to derive a probability model of equilibrium, then finding an SCM that entails that probability model, is unexplored in CCMs. We believe this would be a fruitful avenue for future work.

### 2.6 Proof the inverse Binomial CDF transform is a monotonic conversion.

**Definition 2.2. Monotonic condition.** A variable $Y$ is said to be monotonic relative to variable $X$ in a given structural causal model if and only if, given $X = x$ and noise variable $N = n$, the structural assignment $f_Y(x, n)$ is monotonic in $x$ for all $n$. If the monotonicity condition is true, then $E(Y|do(X = x)) >= E(Y|do(X = x')) \Rightarrow f_Y(x, n) \geq f_Y(x', n) \,\forall n$ [6, 7].

**Definition 2.3. Monotonic conversion.** A monotonic conversion is a conversion of a probabilistic generative model of $Y$ to a structural assignment (which assigns a value of $Y$ to a deterministic function of a random noise input) such that the assignment satisfies the monotonic condition.

*Lemma* 2.2. Let $N$ be a noise variable with a Uniform$(0, 1)$, and let $n$ be a sample of $N$. Let random variable $Y$ be generated from a probabilistic generative model $Y \sim \text{Binomial}(T, \theta(x))$. Let $T$ be the total number of trials, and $\theta(x)$ be the success probability, where $\{x, \theta(x) : x \in \mathbb{N}, 0 < \theta(x) < 1\}$. Assume that $\theta(x)$ is monotonic in $x$ (as in Eq. (15), except with $X_2$ held constant). Let $F^{-1}(\theta(x), T, n)$ denote the inverse Binomial CDF of $Y$, parameterized by $\theta(x)$ and $T$. If $E(Y|do(X = x)) \geq E(Y|do(X = x'))$, then the inverse CDF of $Y$ $f_Y(x, n) = F^{-1}((\theta(x), T, n))$ is a monotonic conversion.

*Proof.* Let $y = E(Y|do(X = x))$ and $y' = E(Y|do(X = x'))$. Given $y$ and $y'$, there exists a value of $n^* \in (0, 1)$ such that $y = F^{-1}(\theta(x), T, n^*)$ and $y' = F^{-1}(\theta(x'), T, n^*)$. Therefore, if $y \geq y'$ then $F^{-1}(\theta(x), T, n^*) \geq F^{-1}(\theta(x'), T, n^*)$.

$\square$

### 2.7 Poisson distribution example

The examples and case studies in the manuscript each had equilibrium distributions that factored into binomial distributions. However, these results are not specific to the Binomial distribution. The following illustrates a similar model that works with the Poisson distribution. Suppose that $T_y$ in Eq. 11 were unknown. Within the probabilistic modeling framework, we can model $T_y$ as a latent variable with distribution $\pi_T$.

$$T_y \sim \pi_T \tag{12}$$

$$\theta_y(X_1, X_2) = \frac{v_1 X_1}{v_1 X_1 + v_2 X_2} \tag{13}$$

$$Y \sim \text{Binomial}\left(T_y, \theta_y(X_1, X_2)\right) \tag{14}$$

A useful result from hierarchical Bayesian modeling is to set $\pi_T$ to a Poisson distribution with parameter $\lambda$, which simplifies the model as follows [10]:

$$\theta_y(X_1, X_2) = \frac{v_1 X_1}{v_1 X_1 + v_2 X_2} \tag{15}$$

$$Y \sim \text{Poisson}\left(\lambda \theta_y(X_1, X_2)\right) \tag{16}$$

We can then set the structural assignment for $Y$ in the SCM using an inverse Poisson CDF. Let $F_{\text{Pois}}^{-1}(\mu, n)$ be the inverse CDF transform that given a parameter $\mu$ and a variable $n$ sampled from a uniform on the unit interval, returns a sample of a Poisson-distributed random variable distributed according to a Poisson distribution with mean $\mu$:

$$n \sim \text{Uniform}(0, 1) \tag{17}$$

$$Y \sim F_{\text{Pois}}^{-1}(\lambda \theta_y(X_1, X_2), n) \tag{18}$$

In general, the approach outlined in this manuscript works with any closed-form equilibrium conditional distributions derived from the underlying Markov process model.