[Reviews · NeurIPS 2019]

Reviewer 1



This paper proposes a framework for casting a Markov process model at equilibrium as an SCM to enable counterfactual inference. The authors claim that the proposed procedure alleviates the identifiability drawback of the SCMs. Then the authors show that with the monotonic assumption, the SCM is identifiable. It is not quite clear to me why it is the benefit of the proposed procedure. The proposed procedure assumes the Binomial distribution of the observed variables, which is quite restrictive. It would be much better if the paper could cover more general cases. After rebuttal----- Thanks for the rebuttal. It has solved my confusions. I have raised the score to 6.

Reviewer 2



The key research questions of this paper concern the conditions under which a Markov process model of a system observed at equilibrium can be successfully converted to an SCM. A complete answer to this question would require proofs of soundness (Any input MPM is successfully converted to an SCM) and completeness (All MPMs can be converted to SCMs), which are not included in the paper. In their response, the authors note that "any probability model there exists a class of SCM models that are equivalent to that probability model in distribution," and say that they will "emphasize this point in the revised manuscript by including a lemma that translates this general result to our case." At a technical level, the paper is dense. For complete understanding of this paper, readers would have to understand Markov process models, structural causal models, some amount of molecular biology, and several areas of mathematics. I apparently lack the background in Markov process models, molecular biology, and perhaps some areas of mathematics to fully understand the contributions of the paper. In their response, the authors promise to "adjust the text to make the key findings more succinct, and move biological exposition and mathematical details that are not essential to communicating those findings to Supplementary Materials." The authors position the work well within the larger context of work on causal inference, including some fairly obscure but extremely interesting recent work on causal inference applied to RL.

Reviewer 3



UPDATE afther the authors' rebuttal: thank you for your response to our comments, was interesting to read about the other reviewers' comment as well. I still believe your work is worth presenting at NIPS; now the meta-reviewers have their difficult job to do and make a selection. Best wishes for your work I would love to see published soon. After the review period, I could help reading your manuscript more in-depth (incl. the Suppl material) and found a few remarks for you to fix: - l81+82 notations, I assume it reads ...$ X_i $, $ \mathbf{p} = ... and then $ X_i \tilde p_i(... $, correct? NB you should define the noises N_i here and not on l89. - eqn (4), you gave me a hard time on this one :). According to Wilkinson 2011, p194, the second term (h_2) of the sum on the RHS should be for Y(t)+1 instead of Y(t)-1. Seems to be what you wrote on l69 of your suppl. material, so reinsuring re: what you implemented! - eqn (6) and (7): on a more careful reading, the g is not g(u)=u/(u+1), the derivaiton of the suppl. material reads OK to me so should be \frac{v_1 X_1(t)}{v_1 X_1(t) + v_2 X_2(t)} - l236: shouldn't it be E[K3(t)] instead of K3(t) in the RHS of the equation? This is what seems correct (and written in Eqn (4) of the Suppl. material...which has missing brackets as a typo) The work in this paper describe the merging of Markov processes with SCM, ending up with the benefits of both models. To me, this is a very interesting, solid contribution, clearly worth presenting at NIPS. I list my questions along the text: - SCM in abstract: avoid acronym (and define it first time used) - l18: biochemical kinetic law is ONE possible modelling of a biological system (making a few 'averaging' assumptions btw), others exist. - in the introduction, I would introduce the reference to Wilkinson's book. - l46: to be pedantic, can we 'empirically' demonstrate a statement? You rely on two case studies only so more a proof of concept. - l61: invariantS or invariant components (i.e. \sum_{i=1}^J X_i (t) = constant)? - l71: 'occurs when': is it a iif, or a sufficient condition only? - l80: not obvious that D needs to be acyclic. From a definition point of view, yes, from a purely causal perspective, this is a limiting assumption, could be discussed later? Ideas to extend this? - l81-82: notation X_i p = ... and X_i p_i unclear - l103-104: I assume the 2 distributions under the X_i^* = x do operator are DEFINED as equivalent, if it's a property, I am lost, how do you prove this? - l114: again, some discussion on the approximation quality of plugging in P_N^{C;X=x} for P_N^C ? - l115: the notation looks like a burden, is it needed, you don't seem to use it in your manuscript... - Algorithms 1 and 2: I sympathise you are limited by space, yet the presentation of both algorithms side by side is poor. Quality of the $ \approx $ at the end of algorithm 1, where is it presented/discussed? In algorithm 2, it is 0:ssize or 1:ssize? No other parameters to be passed on to intModel()? - Say that from l144, you depict the Algorithm 1 steps. l145: a reference or a short explanation, as this is not obvious to me? - l150: binomial or Bernoulli? - l158: an interpretation of the inequality? - l163: ...studies 1 and 2 THEREAFTER, it may... - l207-209: noise is assumed to be the same, not the deterministic part (averages), right? - I am sure you knew someone was going to complain: Fig. 1 (b) and Fig. 2 (a) are barely readable when printed, might pay to try something and to make more precise statement in the text about what you get out of these plots. I would like to finish by congratulating the authors. Quite appreciated the 'discussion' type of research and the fact that you provide the code. Very convincing work.

[Author Response · NeurIPS 2019]

We thank the reviewers for the constructive feedback and are happy to provide clarifications.

**To reviewer 1:**

We would like to stress the benefit of this work. The advantage of dynamic models such as Markov Process models
is the ability to predict the effect of interventions on complex systems (e.g., drug interventions). Applying policy
evaluation methods from reinforcement learning to the policy of selecting optimal interventions is useful, as it increases
the efficiency of the process. This manuscript considers policy evaluation for this class of models, and extends it to
counterfactual policy evaluation.

The counterfactual policy evaluation is useful in two ways. First, it saves resources. For example, it gives a drug
company a way to predict the effectiveness of a perturbation using past data collected under a different policy, without
expending resources in collecting data under the proposed policy. Second, predictions anchored on past data are
more robust to model misspecifications than the pure intervention predictions of Markov process models. The revised
manuscript will emphasize these benefits of this approach, will provide a detailed discussion of the counterfactual
offline policy evaluation use case in Supplementary Materials, and will comparing intervention predictions from the
derived SCM with predictions from the original Markov process model under model misspecification.

Reviewer 1 comments that our focus on the Binomial distribution is restrictive, and suggests for coverage of more
general cases. We would like to clarify that the proposed approach works with any closed-form equilibrium conditional
probability distribution. The challenge is in whether it is possible to derive this distribution from a given Markov
Process model. Generally, with enough simplifying assumptions, it possible to derive any conditional probability
distribution. We previously avoided this discussion because the validity of the assumptions depends heavily on the
application domain. However, we agree with the reviewers that we undershot the mark, and will add this discussion to
the manuscript, as well as a more extensive discussion to Supplementary Materials. To be specific, we will include in
Supplementary Materials a demonstration of the approach with a Poisson-distributed conditional probability distribution,
derived from a slight adjustment in the assumptions in our examples.

**To reviewer 2:**

Reviewer 2 advocates for proofs of soundness and completeness. Indeed, we would like to clarify that for any probability
model there exists a class of SCM models that are equivalent to that probability model in distribution [1]. We will
emphasize this point in the revised manuscript by including a lemma that translates this general result to our case. The
lemma will state that if there exists a closed-form equilibrium probability distribution of the Markov Process (MP), then
there exists a class of Structural Causal Models (SCM) that are equivalent to the MP model's equilibrium probability
model both in observations and in interventions. Our monotonicity constraint then selects an SCM model from that
class, in a way that enables the identification of key counterfactual quantities necessary for policy evaluation. As
mentioned in response to Reviewer 1, modelers can apply simplifying assumptions that attain an equilibrium probability
model, subject to the validity of these assumptions within a problem domain.

Reviewer 2 suggests that the audience would benefit from a clearer and less technically dense description of the key
research findings. We take this suggestion to heart and will adjust the text to make the key findings more succinct,
and move biological exposition and mathematical details that are not essential to communicating those findings to
Supplementary Materials.

**To reviewer 3:**

We are grateful to Reviewer 3 for the suggested corrections and clarifications and will fix or clarify each of these points
in the final manuscript. Due to the space constraint of this letter, we answer three of Reviewer 3's broader questions.

Reviewer 3 asks about extensions to cyclic graphs. In this work, we limited our modeling assumptions to the acyclic
graphs. However, we agree that since MP models and SCMs accommodate cycles, this is a logical avenue for future
work. We will discuss this limitation and extension to future research in the discussion section.

Reviewer 3 asks about the equivalence of distributions for $X_i^*$ and $X_i$ under the do-operator. This equivalence is
implicit in the proof in Section 2.4.1 of Supplementary Materials. In response, will provide more explicit answers. We
will also include in Supplementary Materials a discussion of the recent work connecting interventions on ODE models
and SCMs, and its relationship to the proposed method.

Regarding the reviewer's question about assuming the "noise" to be the same across these distributions in lines 207-209,
the answer is yes.

## Footnotes

[1]Peters, Janzing, Schölkopf. *Elements of Causal Inference: Foundations and Learning Algorithms*, MIT Press, 2017


[Meta-Review · NeurIPS 2019]

As pointed out by the reviewers, these are the strengths and weaknesses of the paper: STRENGTHS The paper addresses the problem of converting a continuous-time Markov process model (MPM) to a structural causal model (SCM). The main advantage of such conversion is that it enables counterfactual inference in non-linear dynamic systems. This is demonstrated through two molecular biology case studies. FOR IMPROVEMENT The authors need to improve the presentation significantly, in order to make the paper accessible and readable. Another important point that should be addressed is the soundness and completeness of converting MPM to SCM.